# Family Income Level, Income Structure, and Dietary Imbalance of Elderly Households in Rural China

**DOI:** 10.3390/foods13020190

**Published:** 2024-01-06

**Authors:** Gangyi Wang, Yanzhi Hao, Jiwei Ma

**Affiliations:** College of Economics and Management, Northeast Agricultural University, No. 600 Changjiang Road, Xiangfang District, Harbin 150030, China; awgy@cau.edu.cn (G.W.); haoyanzhi1016@neau.edu.cn (Y.H.)

**Keywords:** family income, income structure, dietary imbalance, negative binomial regression

## Abstract

In rural areas, the aging of households is becoming increasingly severe, and the issue of dietary imbalance among the elderly is becoming increasingly prominent. Using data from the China Health and Nutrition Survey (CHNS), the negative binomial regression method was used to investigate the impact of household income level on dietary imbalance among rural elderly people, and to explore the heterogeneity of household income structure and its role in the relationship between the two. Research has found that an increase in total household income significantly improves the dietary quality of rural elderly people, and the income structure variable enhances its negative pulling effect on dietary imbalance. For elderly people with moderate dietary imbalance, the effect of increased family income is most significant. In different income groups, the impact of total household income on dietary imbalance in the high-income group is greater than that in the low-income group, and there is also a significant difference in the role played by the proportion of net income from agricultural operations. Therefore, it is necessary to increase the non working income of the elderly, strengthen social responsibility for elderly care, and alleviate the problem of dietary imbalance among rural elderly people.

## 1. Introduction

In the past few decades, the income of Chinese residents has grown rapidly, the income structure has continuously improved, and the dietary structure and quality of urban and rural residents have also undergone significant changes [1,2]. However, due to the long-standing problem of income and consumption stratification between urban and rural areas [3], rural areas generally lag behind urban areas in terms of the diversified food supply, food availability, and balanced nutrient intake [4,5]. In 2021, the global population of elderly people aged 65 and above was 761 million, and exceeded 200 million in China. More than 90% of elderly people in China suffer from dietary imbalances, and over 78% suffer from at least one chronic nutrition-related disease. The number of disabled elderly people is also increasing. The proportion of elderly people and households in rural China was 17.72% and 35.38%, respectively, which were 6.61 and 13.52 percentage points higher than those in urban areas. The phenomenon of the urban–rural inversion of population aging in China is becoming increasingly severe, and the “growing inequality” is seriously reflected in the food access, dietary structure, and dietary quality of rural elderly people. A reasonable diet and balanced nutrition are the foundation of physical health for elderly people in rural areas. However, rural elderly people generally suffer from insufficient food intake and low dietary quality, which leads to increasingly prominent health issues related to nutrition. According to monitoring data on the nutrition and health status of Chinese residents, the malnutrition rate and anemia rate of elderly people aged 75 and above are 10.1% and 17%, respectively, while in rural areas, these two rates are as high as 13.4% and 20%.

It is generally recognized that dietary structure is always closely related to population and family structure. At the macro level, the changes in population structure and dietary structure caused by changes in life expectancy are mutually causal [6,7]. At the micro level, the changes in household income structure driven by the core of family structure and the miniaturization of family size are related to the dietary quality and nutritional health of rural elderly people [8,9,10]. Family elderly care is still the most important way of providing for the elderly in contemporary rural China [11], and the impact of family factors on the diet of the elderly cannot be ignored [12]. Family structure usually affects food needs and types, while family size determines the amount of food required. The dietary structure and habits of elderly people are influenced by the preferences and needs of other family members [13]. The larger the family size, the greater the demand for food, which may increase the variety of food and thus improve the quality of the family’s diet [14]. In addition to controlling variables such as family population structure and size, the impact of family income on the dietary quality of rural elderly people is more practical [15]. The relationship between income and dietary nutrition is one of the ongoing research topics in the field of health economics. Numerous studies have shown that changes in income levels can lead to changes in nutrient levels and dietary quality [16], and income growth is beneficial for promoting the nutritional upgrading of rural residents [17]. However, existing research focuses more on the impact of income levels on dietary quality, while neglecting the importance of income structure. Research has found that nonagricultural income is beneficial for improving the dietary quality of rural residents [18], while rural residents whose main source of income is agricultural income have poorer nutritional status [19,20]. Therefore, how to enhance the family and income effects, improve the dietary quality of rural elderly people, and provide nutritional protection for the vulnerable group of rural elderly people is of great significance for implementing the national strategy of actively responding to population aging [21].

In summary, based on four CHNS data periods from 2004 to 2011, this article evaluates the dietary quality of rural elderly people using DBI-16 and reflects the household income structure of rural elderly people using the proportion of agricultural operating income and the proportion of private net transfer payments. On this basis, the focus is on analyzing how household income level and income structure affect the dietary imbalance of rural elderly people, as well as the role of income structure variables in the relationship between total household income and dietary imbalance.

## 2. Data and Methodology

### 2.1. Data Source

The data used in this article come from the China Health and Nutrition Survey (CHNS), which covers 9 provinces in China with significant differences in geography, economic development, public resources, and health status, covering detailed information such as demographic statistics, work income, lifestyle, and food consumption. Given the availability of data, this article constructed a dataset based on four surveys conducted in 2004, 2006, 2009, and 2011. The research object of this article is the elderly population aged 65 and above living in rural areas. After removing missing and outliers from the data, a total of 3614 valid samples were obtained, including 812 in 2004, 900 in 2006, 1087 in 2009, and 815 in 2011.

### 2.2. Model Settings

The explanatory variable of this study is the dietary quality distance (discrete nonnegative integer), and using the least squares method (OLS) for estimation may lead to heteroscedasticity issues. In addition, the variance of dietary quality distance in the sample data is higher than its mean, indicating a transitional dispersion of dietary quality distance. Therefore, this article chooses a negative binomial regression model for empirical testing.

The probability density function of the negative binomial regression model is Equation (1):(1)$$P_{r}\left(Y_{it}=n\;|\lambda_{it},\alpha \right)=\frac{\Gamma \left(y_{it}+\alpha \right)}{\Gamma \left(y_{it}+1\right)\Gamma \left(\alpha \right)}{\left(\frac{\alpha }{\alpha +\lambda_{it}}\right)}^{\alpha }{\left(\frac{\lambda_{it}}{\alpha +\lambda_{it}}\right)}^{y_{it}},n=0,1,2,3\cdots$$
where $Y_{it}$ is the dependent variable, $\lambda_{it}$ is the expected value of the sample distribution, α is the overdispersion parameter, and $\Gamma \left(\cdot \right)$ is the gamma distribution function. To test the impact of total household income on dietary imbalance among rural elderly people, a regression model was constructed (2):(2)$$Y_{it}=\beta_{0}+\beta_{1}Lnhinc_{i,t}+\lambda Controls_{i,t}+\delta province_{i}+\varepsilon year_{i}+\varepsilon_{i,t}$$

To examine the impact of different household income structures on dietary imbalance among rural elderly people, a regression model was constructed (3):(3)$$Y_{it}=\beta_{0}+\beta_{1}Lnhinc_{i,t}+\beta_{1}Agri_{i,t}+\beta_{1}Pritrans_{i,t}+\lambda Controls_{i,t}+\delta province_{i}+\varepsilon year_{i}+\varepsilon_{i,t}$$

In models (2) and (3), $Lnhinc_{i,t}$ is the logarithm of individual i’s total household income in period t; $Agri_{i,t}$ represents the proportion of net income from agricultural operations to total income; $Pritrans_{i,t}$ represents the proportion of private net transfer payments to total income; $control_{i,t}$ $province_{i}$ is the general term for each control variable; $year_{i}$ is a provincial dummy variable, is a year dummy variable, used to control for time fixed effects and regional fixed effects; and $\varepsilon_{i,t}$ is a random error term.

### 2.3. Variable Description

#### 2.3.1. Dependent Variable

In this study, we used the dietary quality distance (DQD) proposed in the Chinese Dietary Balance Index 16 (DBI-16) revised by He Yuna et al. (2018) [22] to evaluate the dietary imbalance of rural elderly people. DBI-16 is based on the 2016 Chinese Dietary Guidelines and the Revised Balanced Diet Pagoda, consisting of 8 indicators, namely grains, vegetables and fruits, milk and soybeans, animal-based foods, pure energy foods, seasonings, food types, and water. When each indicator reaches the recommended amount, the value is 0; otherwise, the values are negative (−12 to −1) and positive (1 to 12) to evaluate the levels of insufficient and excessive food in in these food groups. The DBI-16 score includes a high bound score (HBS), low bound score (LBS), total score (TS) and diet quality distance (DQD). The evaluation of the score is as follows: 0 is good, less than 20% of the total score is more suitable, 20–40% of the total score is low, 40–60% of the total score is moderate, and higher than 60% of the total score is high. DQD is the absolute sum of various scores, reflecting the average level of overall dietary quality, with a score range of 0–96 points. A DQD score of 0 indicates that there is neither insufficient nor excessive intake in the diet; a score of 1–19 is considered appropriate; a score of 20–38 is considered low-degree dietary imbalance; a score of 39–57 is considered moderate dietary imbalance; and a score above 57 is considered high-degree dietary imbalance.

#### 2.3.2. Independent Variable

Total household income: According to the nature of income sources, the total household income of rural elderly people can be divided into four parts: net operating income, wage income, net property income, and net transfer income [23]. Among them, net operating income can be divided into agricultural operating income and non-agricultural operating income. The net transfer income can be divided into government transfer payments and private net transfer payments, which refer to the net transfer payments made by children and other relatives and friends to the elderly. The urban–rural dual structure is the most obvious feature of China’s elderly income security system, resulting in completely different ways of providing for the elderly in urban and rural areas. Most rural elderly people do not have the opportunity to retire, but continue to work with age, without a fixed income, and have to continue engaging in agricultural operations. In 2022, the actual pension level for rural elderly people was only 5.26% of that for urban employees, and their main source of livelihood was their own labor income and economic support from their children [24]. Based on this, to reflect the household income structure of rural elderly people, the model also includes the proportion of net income from agricultural operations to total household income (Agri) and the proportion of net private transfer payments to total income (Prtrans).

#### 2.3.3. Control Variables

Control variables include individual characteristic variables and family characteristic variables. Among them, individual characteristic variables include gender, age, BMI, dietary knowledge and activity level, and whether there is medical insurance. In the CHNS questionnaire, dietary knowledge and activity levels include 7 correct descriptions of vegetables and fruits, food types, staple foods, fats, dairy and soy foods, and physical activities, as well as 5 incorrect descriptions of sugar, high-fat foods, animal foods, physical exercise, and weight. The survey respondents have 6 options to express their opinions: “strongly disagree, disagree, neutral, agree, strongly agree, or do not know”. This article refers to the research by Zhou et al. [25], and judges the respondents’ answers to dietary knowledge and activity perspectives as “correct” and “incorrect”. Correct answers are scored 1 point, while incorrect answers are not scored. Therefore, the range of the dietary knowledge level is “0–12”. The reliability and validity analysis of the questionnaire showed that Cronbach’s alpha was 0.821, the KMO value was 0.887, and the Bartlett test was *p* < 0.001, indicating good internal consistency of the questionnaire data, which meets the requirements of reliability testing and is suitable for factor analysis.

Family characteristic variables include family size, family structure, and the highest level of education in the family. Home-based elderly care has always been in a fundamental position in rural areas [26]. According to the 2014 China Elderly Social Tracking Survey data, up to 82.1% of rural elderly people choose to “Raise children to care for you when you get old.” [27]. Therefore, this article reflects the family structure of rural elderly people based on whether they live with their children or not. Educated family members usually know the role of a balanced diet in achieving health and pay more attention to the dietary mix of the family. Therefore, the highest level of education in the family is selected as the family characteristic variable. At the same time, year dummy variables and province dummy variables are introduced to control the bias impact of regional and time feature factors on the estimation results. Table 1 provides basic descriptive statistics for these variables.

#### 2.3.4. Dietary Quality of Rural Elderly Population

Dietary quality is an evaluation of the health of dietary patterns, aiming at the lasting improvement of individual and overall health [28]. Studies have found that the excessive and insufficient dietary intake of the rural elderly exist at the same time, but the problems of dietary imbalance and insufficient dietary in are more serious [29]. Unhealthy dietary structure may lead to malnutrition, physical function decline and susceptibility to chronic diseases in the rural elderly, thus reducing their quality of life.

The distribution results of DBI-16 scores of various food intakes of the rural elderly are as shown in Table 2, and 68.2% of the rural elderly consume more cereals than the recommended amount (score ≥ 2). Most of the elderly have a good intake of vegetables, soybeans and nuts [0, 2), while the intake of fruits and milk is insufficient (<−2 indicates insufficient intake). The animal food intake of the rural elderly showed the characteristics of insufficient and excessive intake, of which 32.8% of the rural elderly had excessive intake of livestock and poultry meat, and 51.0% and 48.5% of the rural elderly had insufficient intake of aquatic products and eggs. The elderly in rural areas had excessive intake of cooking oil and salt, while the intake of alcohol and sugar was better. Only 29.1% of the elderly in rural areas had more than 10 kinds of food. Food diversity is the basis of a balanced diet, and the dietary structure of the elderly still needs to be reasonably matched.

The distribution of dietary quality of the rural elderly is shown in Table 3. In general, the dietary imbalance of the elderly in rural areas is serious, and the proportion of the elderly with moderate to high dietary imbalance is as high as 91.82%. Among them, 36.11% of the rural elderly have low-intake deficiencies, while the proportion of middle and high-intake deficiencies is 49.61% and 13.31%, respectively. The proportion of elderly people with inadequate intake is only 0.97%. In terms of excessive dietary intake, low and moderate excessive intake accounted for 40.73% and 38.74%, respectively, while the proportion of elderly people with high excessive intake was only 3.79%. The number of rural elderly people with moderate insufficient intake and excessive intake was the largest, followed by the number of people with low insufficient in and excessive in, and the elderly with balanced diet were almost non-existent.

## 3. Empirical Results

### 3.1. Benchmark Regression Analysis

Considering that the explained variable dietary quality distance is a counting variable, this article uses a negative binomial regression model to analyze the relationship between household income and the dietary quality distance of rural elderly people. The regression results are shown in Table 4. (1) The regression results of Equation (2) were reported to analyze the impact of total household income on dietary quality; Equation (3) introduces household income structure variables based on Equation (2), and the regression results are reported in column (2). By comparing the regression results of two equations, we further analyzed the impact of introducing household income structure variables on the dietary imbalance of rural elderly people. All models controlled for year and province-fixed effects.

The regression results between total household income and dietary quality are shown in column (1) of Table 4. The results show that after controlling for individual and family characteristic variables, it was found that the coefficient of total household income was significantly negative at the 1% level, indicating a negative pulling effect of total household income on dietary quality. This means that with the increase in total household income, the phenomenon of dietary imbalance among rural elderly people decreased, which improved the dietary quality of rural elderly people. The regression results in column (2) indicate that the coefficient between the proportion of agricultural net income and the proportion of private net transfer payments was significantly positive at the 1% level. This means that as the proportion of net income from agricultural operations and private net transfer payments to total household income increases, the dietary imbalance among rural elderly people intensifies, reducing their dietary quality. At the same time, by comparing Equations (2) and (3), it can be found that the addition of household income structure variables increases the negative pulling effect of total household income on dietary quality. It can be seen that household income structure is indeed an important economic variable that affects the dietary quality of rural elderly people.

### 3.2. Robustness Analysis

#### 3.2.1. Robustness Testing

To further verify the effectiveness of the impact of household income level and income structure variables on dietary imbalance, this article uses a robustness test by replacing the dependent variable.

The lower-bound score (LBS) is also a calculation method for DBI-16 [30], used to evaluate the situation of insufficient dietary intake. The lower-bound score (LBS) is the absolute sum of the negative points in various scores, reflecting the degree of insufficient dietary take, with a score range of 0–72 points. An LBS of 0 indicates no insufficient intake in the diet, with a score of 1–14 being more appropriate, a score of 15–29 indicating low intake, a score of 29–43 indicating moderate intake, and a score above 43 indicating high intake. This article uses a negative terminal score as the proxy variable for dietary imbalance, re-estimating the dietary quality of rural elderly people, and conducting regression analysis on models (1) and (2) again. The regression results are shown in Table 5, where the coefficient of the total income of elderly households is significantly negative at the 1% level, while the coefficients of net income from agricultural operations and private net transfer payments are significantly positive at the 1% level. Compared with benchmark regression, the regression results show no substantial changes, indicating that the estimation results have strong robustness.

#### 3.2.2. Endogeneity Testing

In the benchmark model analysis, the total household income and income structure variables have a significant impact on the dietary imbalance of rural elderly people. However, the reverse causal relationship between residents’ dietary nutrition and household income may lead to endogeneity issues [31,32]. To alleviate the endogeneity of household income, this article adopts the instrumental variable method for processing. Research has shown that the average household income of elderly people in the same village has a significant driving effect on the income level and income structure of elderly households, and the average household income of elderly people in the same village is not directly related to the dietary quality of the elderly [33]. Therefore, this article uses the average household income of the elderly in the same village as the instrumental variable of total household income. To ensure that the instrumental variables meet the requirements of homogeneity and correlation, this article uses the 2SLS method for testing.

The first-stage regression results show that the impact of the average household income of the elderly in the same village on the total household income of the elderly is significantly positive at the 1% level, indicating that the instrumental variable satisfies the correlation hypothesis. At the same time, the critical value of the Stock-Yogo weak ID test critical values: 10% maximal IV size 16.38, which is much smaller than the minimum eigenvalue statistic 1035.56, indicating the absence of weak instrumental variables, proving that “average household income of elderly people in the same village” is a suitable instrumental variable. The two-stage regression results passed the Hausman test for endogeneity, and at the 1% significance level, there is still a significant negative correlation between the total income of elderly households and their dietary quality. This means that the impact of elderly household income level on their dietary imbalance is robust.

### 3.3. Heterogeneity Analysis

#### 3.3.1. Heterogeneous Effects on the Degree of Dietary Imbalance

In rural areas, the phenomenon of dietary imbalance among elderly people is relatively serious. In the sample data, the most suitable scenario for dietary quality is 0, with 74.60% of elderly people experiencing moderate dietary quality imbalances, while only 9.08% and 16.33% of elderly people experience low and high dietary quality imbalances. This section investigates whether the income level and income structure of rural elderly households have heterogeneous effects on the grouping of dietary quality. Table 6 shows that compared to other dietary quality groups, the total household income and income structure have a greater impact on the rural elderly in the moderate dietary imbalance group. From column (3), it can be seen that in the group with moderate dietary imbalance, without introducing household income structure variables, total household income has a negative pulling effect on dietary quality at the 1% level. After introducing the variable of household income structure, the negative pulling effect of total household income on dietary quality increased, still significant at the 1% level. The proportion of net income from agricultural operations to dietary quality has a positive driving effect at the 1% level, while the proportion of private net transfer payments has no significant impact on the dietary quality of rural elderly people. This means that among the moderately imbalanced population, with the increase in total household income, the dietary imbalance of rural elderly people improved. After considering the income structure, the improvement effect of total household income on dietary imbalance was enhanced. The higher the proportion of net income from agricultural operations, the more severe the dietary imbalance among the elderly.

#### 3.3.2. Analysis of Dietary Quality Estimation Results for Different Income Groups

The impact of total household income on dietary imbalance among rural elderly may have heterogeneity among groups. This article conducted group regression for different income levels, dividing the total household income into two income groups: “below average” and “above average” for testing, as shown in Table 7. The regression results indicate that in the above-average group, the improvement in total household income on dietary imbalance was better than that in the below-average group. Specifically, in the above-average group, the addition of household income structure variables led to a decrease in the improvement effect of total household income on dietary imbalance. As the proportion of net income from agricultural operations increases, the dietary quality of the elderly tends to be imbalanced, and the proportion of private net transfer payments is not related to dietary imbalance. In the below-average group, after adding household income structure variables, the improvement effect of total household income on dietary imbalance was enhanced, and the impact of the proportion of net agricultural income and private net transfer payments on dietary imbalance in the elderly remained consistent with before the grouping.

## 4. Discussion

Dietary quality is an evaluation of the health level of dietary patterns, aimed at persistently improving individual and overall health [28]. In recent years, dietary quality evaluation has also become a hot topic of research both domestically and internationally. The Dietary Quality Index (DQI) is the first comprehensive evaluation index for dietary quality based on food and nutrients established in the United States in 1994 [34]. The Healthy Eating Index (HEI) is currently the most widely used evaluation method, designed in 1995 according to the requirements of the Dietary Guidelines for Americans (DGA) and updated every five years together with the DGA [35]. On the basis of HEI, foreign scholars have also proposed various dietary quality indices, such as the Alternative Healthy Diet Index (AHEI-2010) [36] and the Overall Dietary Index (ODI) [37]. In response to the ideal dietary patterns and dietary habits of Chinese people, domestic scholars have successively proposed the China Dietary Balance Index (DBI) [38] and the China Healthy Diet Index (CHEI) [39] to evaluate the dietary quality and nutritional level of Chinese people. Among them, the Chinese Dietary Balance Index (DBI) was established by He Yuna et al. in 2005 and revised to DBI-16 in 2018 based on the Dietary Guidelines for Chinese Residents (DGC-2016) and the requirements of the Balanced Diet Pagoda [22]. The DBI-16 score includes higher-bound score (HBS), lower-bound score (LBS), and diet quality distance (DQD), which can simultaneously reflect the situation of excessive, insufficient, and imbalanced dietary in. Many scholars at home and abroad have used DBI-16 to evaluate the dietary quality of elderly people in rural China [40,41]. Previous studies have found that both excessive and insufficient dietary intake coexist among elderly people in rural areas, but the problems of dietary imbalance and insufficient intake are more severe [29]. In 2021, 95.22% of elderly people in a rural area of Chengdu, China who participated in community physical examinations had insufficient dietary in, and the proportion of elderly people in a state of dietary imbalance was as high as 99.69% [42]. In a survey of dietary data for elderly people in Shanghai, China, it was found that over 90% of the elderly were rated as imbalanced in negative scores and dietary quality distance [43].

As age increases, changes in physical function, preferences, and income levels will all have an impact on the dietary structure of elderly people [44,45,46], and long-term unreasonable dietary structure will exacerbate the dietary imbalance and insufficient intake of rural elderly people [47]. Therefore, how to improve the dietary quality of elderly people in rural areas has become a focus of relevant research. Existing research mainly focuses on household income, urbanization, population structure, dietary nutrition knowledge, and other aspects. For example, Zhong et al. (2012) found that as the population ages, there are significant changes in food consumption patterns, particularly in the demand for certain types of food [48]. Agbozo et al. (2018) proposed that despite having some knowledge about nutrition, the dietary patterns of these older individuals are not ideal, leading to poor nutritional status [49]. In Tanzania, Cockx et al. (2018) investigated the impact of urbanization on dietary changes among rural-to-urban migrants. They observe a shift from traditional to more Westernized diets among this population. This shift is associated with significant changes in dietary quality, potentially leading to negative health outcomes [50]. The relationship between healthy food choices and ecologically conscious consumer behavior has also been widely studied [51]. Cramarenco et al. (2023) conducted a bibliometric analysis, and found that organic food is perceived as safer and more nutritious than conventionally produced food [52].

Research has shown that income has a positive impact on dietary diversity, and the increase in household income level is related to improvement in dietary quality [53]. Family income directly affects the types and quality of food available to rural elderly people, as well as their food consumption patterns and dietary structure [54]. Engel’s law and Bennett’s law indicate that as household income increases, residents’ diets tend to diversify, and the consumption of starch-based foods gradually decreases, while the consumption of more nutritious foods such as meat, milk, vegetables, and fruits gradually increases [55,56]. Numerous studies have confirmed that changes in income levels can lead to changes in nutritional intake levels and dietary quality. There is a significant negative correlation between poverty and dietary quality. A low income increases the additional restrictions on the elderly to achieve nutritional needs. The nutritional intake and dietary quality of the elderly in low-income rural areas are poor, and there are fewer resources to support overall health [57,58]. In addition to income level, family income structure is also an important factor affecting the dietary nutrition of rural elderly people. However, not all sources of income are equally effective in improving dietary quality [59]. According to the Mental Accounting Theory proposed by Thaler, consumers will divide their income into different accounts, which can be understood as “Earmarking” [60]. The marginal propensity to consume varies among different sources of income, which means that even if the amount of income is fixed, there will be significant differences in its impact on various consumption intensities [61]. Research has shown that household operating income is the most important factor affecting various consumption expenditures of rural residents, while transfer income mainly affects the clothing, food, housing, and transportation of rural residents [62,63]. This provides great reference significance for the selection of family structure variables in this article.

### Strengths and Limitations of This Study

Due to the availability of sample data, this article has many shortcomings. The article does not reflect the gradual changes and mutations that income adjustments, family changes, and disease shocks may cause in the dietary quality of rural elderly people. Firstly, the research object of this study is the national rural elderly dietary health research. However, due to the availability of data, the research data used in this study are CHNS micro-survey data from 2004 to 2011. Although the data can demonstrate the relationship and effect between household income and dietary quality in this paper, the applicability of the empirical research results still needs to be tested by updated data. Therefore, future research should widely collect micro-survey data to improve the accuracy of research conclusions. Secondly, we used the latest data on the household income of the elderly to determine the mechanism of its impact on dietary quality. It does not take into account that the family income of the elderly may change with events such as widowhood and marriage of children. It is necessary to further study the impact of changes in family income on the quality of bone diets of the rural elderly. Lastly, in China, the difference between the rural elderly and the urban elderly is that they do not have retirement wages. Therefore, pension insurance and medical insurance play a vital role in ensuring the dietary quality of the elderly. Future research should consider setting the insurance status of the rural elderly as a control variable to highlight the marginal contribution of family income to the dietary quality of the rural elderly. Further exploration is needed in future research.

## 5. Conclusions

This article was based on the data from four Chinese nutrition and health surveys from 2004 to 2011 and compared and analyzed the impact of the total household income of rural elderly people on their dietary imbalance before and after adding income structure variables. The following conclusions were drawn.

Firstly, the increase in total household income has significantly improved the dietary imbalance of rural elderly people. After considering the income structure, the improvement effect of total household income is enhanced. Secondly, the relationship between insufficient dietary intake and total household income is consistent with the relationship between dietary imbalance and income, which means that the dietary imbalance of rural elderly people is mainly caused by insufficient intake. Thirdly, 74.60% of rural elderly people have a moderate dietary imbalance, and the higher the proportion of net income from agricultural operations, the more severe the dietary imbalance. Fourthly, there is heterogeneity in the improvement effect of total household income. The improvement effect on dietary quality in the high-income group is better than that in the low-income group, and the role of income structure in the two groups is opposite.

## 6. Recommendations

In summary, the conclusion of this article reflects that an increase in total household income is beneficial for improving the dietary quality of rural elderly people. It was revealed that the essence of dietary imbalance among rural elderly people lies in insufficient dietary intake, that it is necessary to strengthen social responsibility for elderly care, moderately increase the nonlabor income of rural elderly people, and ensure good dietary quality; and that with the aging of households, more attention should be paid to the internal differences of their groups, and “tailored medicine” should be given. Specific recommendations are as follows.

It is necessary to increase the economic support of the rural elderly through various channels, so as to improve the food purchasing ability of the rural elderly and promote the improvement of the nutritional status of the rural elderly. First of all, strengthen the construction of the old-age security system, and improve the social assistance and social welfare system on the basis of improving the basic old-age insurance system. For rural areas, especially the elderly with economic difficulties, certain pension service subsidies should be given. Secondly, the main way of providing for the aged in China is still family support. It provides more services and policy cooperation for the family to care of both development and care functions. Children should be promoted to live nearby or together with economic compensation, tax relief and so on. Finally, family members provide companionship to the elderly, but also affect their living habits and dietary knowledge. It can protect the health of the elderly and improve the nutritional literacy of the rural elderly. The community should actively carry out health promotion activities for the rural elderly, and call on the rural elderly to learn nutrition knowledge and improve the quality of their diet.

## Figures and Tables

**Table 1 foods-13-00190-t001:** Descriptive statistics of variables.

Variable	Variable Definition and Assignment	Mean	Std. Dev.
DQD	The higher the score, the more severe the dietary imbalance	49.55	7.95
	DQD = 0, There is neither insufficient nor excessive in in the diet	-	-
	DQD = {1–19}, the diet is more suitable	-	-
	DQD = {20–38}, low degree dietary imbalance	34.90	3.03
	DQD = {39–56}, moderate dietary imbalance	48.77	4.97
	DQD = {57–96}, high dietary imbalance	61.26	3.11
Lnlinc	The logarithm of the total household income in yuan from the previous year	9.07	1.88
Agri	The proportion of agricultural operations income to total household income	0.33	0.38
Pritrans	The proportion of private net transfer payments to total household income	0.19	0.31
GENDER	Male = 1; Female = 0	0.47	0.50
AGE	Unit: Year	72.32	5.82
BMI	Weight/Height^2^ (Unit: kg/m^2^)	22.65	3.86
DK	The higher the assigned value, the higher the level of dietary knowledge and activity	49.56	7.95
MED_insur	Having medical insurance = 1; No medical insurance = 0	6.00	3.31
FAM_stru	Living with children = 1; Not living with children = 0	0.75	0.43
FAM_size	Number of family members (unit: number)	3.86	1.57
MAX_edu	Maximum education years among family members (in years)	6.59	3.96

**Table 2 foods-13-00190-t002:** Rural elderly population distribution of each DBI-16 index segment (%).

Score	Cereal	Vegetables	Fruits	Milk	Beans	Meat	Aquatic	Eggs	Oil	Alcohol	Sugar	Salt	Food Types
−12~(−11)	2.0												
−10~(−9)	0.7												12.9
−8~(−7)	1.7												19.6
−6~(−5)	2.4	10.7	46.1	88.6	34.6								19.8
−4~(−3)	6.4	21.4	48.4	7.5	2.8	6.8	51.0	48.5					18.6
−2~(−1)	7.7	44.7	4.7	3.2	8.8	18.2	10.6	13.0					13.5
0	10.8	23.2	0.7	0.7	53.8	23.8	38.3	8.3	38.6	89.0	96.7	6.7	15.6
1~2	12.0					18.5		18.6	22.8	6.3	1.0	13.2	
3~4	9.8					32.8		11.7	20.9	2.2	1.0	44.3	
5~6	8.4								17.7	2.4	1.4	35.6	
7~8	6.7												
9~10	4.6												
11~12	26.7												

Data source: Statistical analysis based on data from China Health and Nutrition Survey in 2004, 2006, 2009 and 2011.

**Table 3 foods-13-00190-t003:** Distribution of dietary status of rural elderly population.

	Indicator	Range	$\bar{x}\pm s$	Dietary Quality Distribution/%				
				Balance	More Suitable	Low	Moderate	High
Insufficient intake	LBS	0~60	32.9 ± 9.0	0	0.97	36.11	49.61	13.31
Excessive in	HBS	0~44	16.7 ± 6.5	0	16.74	40.73	38.74	3.79
Dietary imbalance	DQD	0~84	49.6 ± 7.9	0	0	9.08	74.60	16.33

Data source: Statistical analysis based on data from China Health and Nutrition Survey in 2004, 2006, 2009 and 2011.

**Table 4 foods-13-00190-t004:** Baseline regression results.

Variable	DQD	
	(1)	(2)
Lnlinc	−0.0082 ***	−0.0084 ***
	(0.0013)	(0.0013)
Agri		0.0572 ***
		(0.0070)
Pritrans		0.0192 **
		(0.0084)
GENDER	−0.0194 ***	−0.0196 ***
	(0.0049)	(0.0049)
AGE	0.0004	0.0009 **
	(0.0004)	(0.0004)
BMI	−0.0035 ***	−0.0023 ***
	(0.0007)	(0.0007)
DK	−0.0050 ***	−0.0039 ***
	(0.0010)	(0.0010)
MED_insur	−0.0193 ***	−0.0023 **
	(0.0074)	(0.0007)
FAM_stru	0.0385 ***	0.0359 ***
	(0.0087)	(0.0087)
FAM_size	−0.0049 **	−0.0058 **
	(0.0024)	(0.0024)
MAX_edu	−0.0034 ***	−0.0023 ***
	(0.0007)	(0.0007)
Province Effect	Control	Control
Year Effect	Control	Control
Constant	4.1092 ***	4.0290 ***
	(0.0410)	(0.0419)
Log likelihood	−12,229.897	−12,196.27
Prob > chi^2^	0.0000	0.0000
N	3614	3614

Note: ** *p*  <  0.05, *** *p*  <  0.001. The data in parentheses represent the robust standard error, and the following text is the same.

**Table 5 foods-13-00190-t005:** Robustness test results.

Variable		Replace the Dependent Variable	2SLS	
	LBS		One-Stage Lnlinc	Two-Stage DQD
Ave-Lnlinc			0.9476 *** (0.0384)	
Lnlinc	−0.0153 *** (0.0018)	−0.0157 *** (0.0018)	-	−0.7850 *** (0.1937)
Agri		0.0858 *** (0.0090)		
Pritrans		0.0310 *** (0.0113)		
Control variables	Control	Control	Control	Control
Province Effect	Control	Control	Control	Control
Year Effect	Control	Control	Control	Control
F statistics			127.17 ***	-
R^2^	0.1004	0.1038	0.2797	0.0764
Hausman test(P > chi^2^)			-	0.0000
Minimum eigenvalue statistic			610.38 > (16.38)	-

Note: *** *p*  <  0.001.

**Table 6 foods-13-00190-t006:** The impact of total household income and income structure on the grouping of dietary quality.

Variables	Low dietary Imbalance		Moderate dietary Imbalance		High Dietary Imbalance	
	(1)	(2)	(3)	(4)	(5)	(6)
Lnlinc	−0.0036	−0.0033	−0.0031 ***	−0.0032 ***	0.0001	−0.0001
	(0.0027)	(0.0028)	(0.0010)	(0.0010)	(0.0011)	(0.0011)
Agri		0.0181		0.0204 ***		0.0051
		(0.0150)		(0.0056)		(0.0065)
Pritrans		0.0153		0.0059		0.0008
		(0.0165)		(0.0065)		(0.0073)
Control variables	control	control	control	control	control	control
Province Effect	control	control	control	control	control	control
Year Effect	control	control	control	control	control	control
Constant	3.6548 ***	3.6350 ***	3.9687 ***	3.9403 ***	4.0782 ***	4.0714 ***
	(0.0853)	(0.0857)	(0.0315)	(0.0324)	(0.0364)	(0.0391)
Loglikelihood	−924.17	−923.94	−8330.36	−8327.28	−1798.66	−1798.60
Prob > chi^2^	0.0035	0.0029	0.0000	0.0000	0.0001	0.0002
N	328	328	2696	2696	590	590

Note: *** *p*  <  0.001.

**Table 7 foods-13-00190-t007:** Impact of different household income levels on dietary quality.

Variables	Below Average Income		Above Average Income	
	(1)	(2)	(3)	(4)
Lnlinc	−0.0052 ***	−0.0082 ***	−0.0210 ***	−0.0121 **
	(0.0017)	(0.0018)	(0.0054)	(0.0060)
Agri		0.0534 ***		0.0576 ***
		(0.0114)		(0.0105)
Pritrans		0.0216 *		−0.0013
		(0.0114)		(0.0248)
Control variables	control	control	control	control
Province Effect	control	control	control	control
Year Effect	control	control	control	control
Constant	4.1413 ***	4.0816 ***	4.0947 ***	4.0292 ***
	(0.0618)	(0.0628)	(0.0601)	(0.0780)
Loglikelihood	−4952.97	−4942.82	−7240.83	−7224.77
Prob > chi^2^	0.0000	0.0000	0.0000	0.0000
N	1484	1484	2130	2130

Note: * *p*  <  0.10, ** *p*  <  0.05, *** *p*  <  0.001.

## Data Availability

Data is contained within the article.

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
