# Peer review of "Family Income Level, Income Structure, and Dietary Imbalance of Elderly Households in Rural China"

_foods, 2024, doi:10.3390/foods13020190_

Round 1
Reviewer 1 Report
Comments and Suggestions for Authors
The manuscript with the title “Family income level, income structure, and dietary imbalance of elderly households in rural China” provides an analysis of dietary imbalance among elderly from rural China, with special attention to factors. The manuscript is insightful and well-written.
I suggest authors to provide a few original recommendations/suggestions or approaches towards improving the dietary quality of elderly people and reduce the imbalance in rural areas.
Best regards.
Comments on the Quality of English Languageoverall English quality is OK, but some syntax and style adjustments are needed.
Author Response
Dear Reviewer:
I sincerely thank you for your valuable suggestions, which are very academically valuable. My team and I have carefully read these suggestions and rewrite the introduction and theoretical framework of the paper according to the revision suggestions, based on careful consideration and extended reading of the literature.
Please see the attachment.

Reviewer 2 Report
Comments and Suggestions for Authors
In the abstract, kindly check the full meaning of the acronym "CHNS". It is not the same with the one in the he main text.
From the introduction, it is important that authors justify the study focus on rural elderly Chinese. There is a need for the authors to present a more robust background information on this study too.
In line 32-33, what is the age range of the group called "elderly people". You expected to present the current data of global elderly and that of china. Give detail recent food and nutrition statistics of the group.
In equation 1, Yit, is the dependent variable called?
Where is the result of the dietary quality of the elderly people?
You expected to present it before the regression model analyses. Only the mean and standard deviation are insufficient. Present a detailed results on dietary quality/imbalance among the 3,614 rural elderly Chinese.
However, before the conclusion section, kindly present a sub-section with the heading "limitations of the study" and "areas for further research"
Thank you.
Comments on the Quality of English LanguageFine
Author Response

(The authors gave the same response as above.)

Reviewer 3 Report
Comments and Suggestions for Authors
‘In the past few decades, the income of Chinese residents has grown rapidly, the income structure has continuously improved, and the dietary structure and quality of urban and rural residents have also undergone significant changes’ – this idea is valid for so many countries, try and be specific by providing concrete info. The paper would benefit from a clearer research question or argument around which it could be more clearly structured. ‘the Urban-rural inversion of Population Aging in’ – why initial word capitalization? "growing inequality", "transitional dispersion", "retire", "take care of children and prevent aging", "tailored medicine" – why in quotes? More development and depth of the methodology and analysis are needed. ‘More noteworthy is that’ – avoid such phrases, the ideas must speak for themselves. Try and provide more references to support your ideas that are typically substantiated by only one source - and as recent & relevant as possible. ‘based on four CHNS data periods from 2004 to 2011’ – way too old data, integrate as recent as possible. How could they cumulatively reflect the current picture of the topic? You should compare your results with others in terms of concrete data for better research integrative value. ‘Research has shown that the average household income of elderly people in the same village has a significant driving effect on the income level and income structure of elderly households, and the average household income of elderly people in the same village is not directly related to the dietary quality of the elderly.’ – what specific research? The paper requires revisions to contextualize the merits of the study and potential uses of its methodology in future studies. Several sentences provide close to zero info: ‘In recent years, dietary quality evaluation has also become a hot topic of research both domestically and internationally [32]’. ‘Existing research mainly focuses on household income, urbanization, population structure, dietary nutrition knowledge, and other aspects [50–53]’. ‘The impact of both on the dietary imbalance of elderly people in rural areas should be the focus of this article. Therefore, it is necessary to explore the impact of household income level and income structure on the dietary quality of rural elderly people.’ – ‘should be’, ‘explore’ – ‘is’, ‘was necessary to explore’. Some bibliographic references are simply brought up without being developed, or without an adequate explanation as to why they are relevant. Conclusion needs to be rewritten so that only important results are brought out along with their interpretation, comparison with earlier studies, and implications in a more integrated fashion. There is some discussion of the limitations of the study however these are not considered in terms of the implications on the study findings. Where can future research further contribute to this area and what are the possible implications of this research on literature, practice and policy? The reference list is not properly edited.
The relationship between healthy food choices and ecologically conscious consumer behavior as regards family income level, income structure, and dietary imbalance of elderly households, and thus such sources can be cited:
Majerova, J.; Sroka, W.; Krizanova, A.; Gajanova, L.; Lazaroiu, G.; Nadanyiova, M. Sustainable Brand Management of Alimentary Goods. Sustainability 2020, 12, 556. https://doi.org/10.3390/su12020556
Cramarenco, R.E., Burcă-Voicu, M.I., Dabija, D.C. 2023. Organic food consumption during the COVID-19 pandemic. A bibliometric analysis and research review. Amfiteatru Economic, 25(Special Issue 17), pp.1042-1063. https://doi.org/10.24818/EA/2023/S17/1042.
Author Response

(The authors gave the same response as above.)

Round 2
Reviewer 2 Report
Comments and Suggestions for Authors
Thank you for send the revised version of this manuscript. The editor can make final publication decision on this revised version. Thank you.
Comments on the Quality of English LanguageFine
Reviewer 3 Report
Comments and Suggestions for Authors
This revised version can be published.